# When Are Solutions Connected in Deep Networks?

**Quynh Nguyen**
MPI-MIS, Germany
quynh.nguyen@mis.mpg.de

**Pierre Bréchet**
MPI-MIS, Germany
pierre.brechet@mis.mpg.de

**Marco Mondelli**
IST Austria
marco.mondelli@ist.ac.at

## Abstract

The question of how and why the phenomenon of mode connectivity occurs in training deep neural networks has gained remarkable attention in the research community. From a theoretical perspective, two possible explanations have been proposed: *(i)* the loss function has connected sublevel sets, and *(ii)* the solutions found by stochastic gradient descent are dropout stable. While these explanations provide insights into the phenomenon, their assumptions are *not* always satisfied in practice. In particular, the first approach requires the network to have one layer with order of $N$ neurons ($N$ being the number of training samples), while the second one requires the loss to be almost invariant after removing half of the neurons at each layer (up to some rescaling of the remaining ones). In this work, we improve both conditions by exploiting the quality of the features at every intermediate layer together with a *milder* over-parameterization requirement. More specifically, we show that: *(i)* under generic assumptions on the features of intermediate layers, it suffices that the last two hidden layers have order of $\sqrt{N}$ neurons, and *(ii)* if subsets of features at each layer are linearly separable, then almost no over-parameterization is needed to show the connectivity. Our experiments confirm that the proposed condition ensures the connectivity of solutions found by stochastic gradient descent, even in settings where the previous requirements do not hold.

## 1 Introduction

The aim of this work is to provide further theoretical insights into the mode connectivity phenomenon which has recently attracted some considerable attention from the research community (i.e. training a neural network via stochastic gradient descent (SGD) from different random initializations often leads to solutions that could be connected by a path of low loss) [2, 9, 11, 12, 15]. Existing theoretical insights from optimization landscape analysis suggest that, for over-parameterized networks, the loss function has connected sublevel sets, and hence the solutions found by SGD are also connected [32, 37, 43]. However, in order for such a result to hold, the network needs to have one wide layer with at least $N + 1$ neurons ($N$ being the number of training samples) [33]. This bound is also known to be tight for 2-layer networks. Here, we would like to understand the above phenomenon for deep and mildly over-parameterized networks (i.e. we aim to prove weaker guarantees under milder conditions). The idea is that, even when the sublevel sets are disconnected, SGD may only concentrate in a (large) region with connected minima, see e.g. Figure 1. This leads to the question that whether one can characterize the properties of the solutions found by SGD which capture the mode connectivity phenomenon, for mildly over-parameterized networks. Along this line, a recent work [24] proposed the so-called "dropout stability" condition, that is, removing half the neurons from every hidden layer leads to almost no change in the value of the loss. As shown in the prior

35th Conference on Neural Information Processing Systems (NeurIPS 2021).

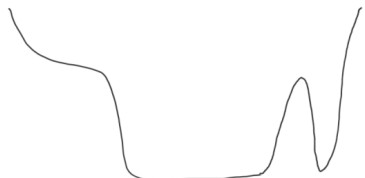

Figure 1: A function with disconnected sublevel sets and two connected sets of minima.

works (also in our experiments in Section 6), this condition holds for very wide networks or when dropout is used as a regularizer during training, but it does not hold under other practical settings.

The aim of this work is to provide a new characterization of SGD solutions which capture their connectivity property, and to improve the previous assumptions (width $\Omega(N)$, and dropout stability) for networks of practical sizes, learned without the support of dropout training. As a remark, although our work is motivated by the original phenomenon concerning mainly the solutions found by SGD, all the theoretical results are still valid for other pairs of solutions in parameter space.

**Main contributions.** Our main Theorem 4.1 provides an upper bound of the loss along a path connecting two given solutions in parameter space. This bound is characterized by the loss of $L$ *optimized and sparse* subnetworks, $L$ being the network depth. These $L$ subnetworks connect subsets of at least half the neurons at each layer to the output (see Figure 2), and their weights can be optimized independently of the weights of the original network, which leads to a significant reduction of the loss. This is the key difference to the previously proposed dropout stability property [24]. Next, we specialize our theorem to different settings to demonstrate the improvement over prior works:

- Corollary 4.2 shows that our condition is provably milder than dropout stability [24]. In fact, dropout stability implies that the loss does not change much when we remove at least half of the neurons at each layer and rescale the remaining ones. Here, we allow to optimize again all the weights of the subnetworks connecting the subsets of features to the output.

- As the weights of the subnetworks can be optimized, we establish a formal connection between mode connectivity and memorization capacity of neural nets. In particular, Corollary 4.3 proves that, if the last two hidden layers have $\Omega(\sqrt{N})$ neurons, then under some additional mild conditions on the features of intermediate layers, any two given solutions can be connected by a path of low loss. This improves the width condition of prior works [32, 33, 43] from $\Omega(N)$ to $\Omega(\sqrt{N})$ at the cost of ensuring a weaker guarantee (i.e. the connectivity of sublevel sets proved in prior works implies the connectivity of any two pairs of solutions, but in this paper we only show the connectivity for certain pairs).

- Corollary 4.4 proves that, for a classification task, if subsets of features at each layer are linearly separable, then no over-parameterization is required to prove the connectivity.

The message of this paper is that, in order to ensure the connectivity, a smooth trade-off between feature quality and over-parameterization suffices. The two extremes of this trade-off are captured by Corollary 4.3 and 4.4 mentioned above. More generally, at each layer $l$, we pick a subset $\mathcal{I}_l$ containing at least half the neurons. If the features indexed by $\mathcal{I}_l$ are good enough, in the sense that there exists a small subnetwork $\xi_l$ achieving small loss when trained on these features as inputs, then all layers $k > l$ in the original network do not need to be larger than twice the widths of the subnet $\xi_l$. Hence, the better the features at lower layers, the less over-parameterized the higher layers need to be.

Finally, our numerical results show that, if the learning task is sufficiently simple (e.g. MNIST) and the network is sufficiently over-parameterized, then dropout stability holds and the SGD solutions can be connected as in [24]. However, when the learning task is more difficult (e.g. CIFAR-10), or for networks of moderate sizes, dropout stability does not hold and the path of [24] exhibits high loss. In this challenging setting, we are still able to find low-loss paths connecting the SGD solutions.

**Proof techniques.** We remark that, while previous work [32, 33, 43] focused on the connectivity of all the sublevel sets (i.e. a property of the global landscape), here we are only interested in the connectivity of a particular subset of solutions, especially those that are discovered by SGD. This leads to several differences in the proofs. First, previous analysis relies on the fact that the first hidden layer has at least $N + 1$ neurons [33], and thus the training inputs can essentially be mapped to a

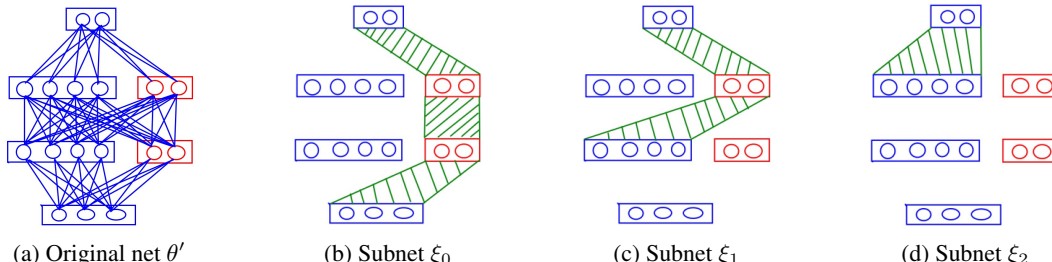

(a) Original net $\theta'$       (b) Subnet $\xi_0$       (c) Subnet $\xi_1$       (d) Subnet $\xi_2$

Figure 2: An illustration of the sub-networks appearing in our main Theorem 4.1. The blue neurons correspond to the subset $\mathcal{I}_l$ in the theorem, and its cardinality is at least half the layer width. Each sub-network $\xi_l$ receives as inputs the blue neurons from layer $l$, and pass it through the red ones at the higher layers, all the way to the output of the network. All the weights of the sub-networks (in green) can be re-optimized to minimize the upper bound in the theorem. This is shown via the infimum taken over $\xi_l$ on the RHS of Equation (4).

feature space where they become linearly independent. This property allows the network to realize arbitrary outputs at each layer, which leads to the connectivity of sublevel sets. This property is no longer true for the networks considered here, where the layer widths are allowed to be *sublinear* in $N$. Second, the analysis of sublevel sets often do not take into account the bias of SGD, whereas here we focus more on understanding the properties of these solutions which lead to their connectivity. More closely related is the work by [24], with which we provide a detailed discussion in Section 4.

## 2 Further Related Work

Theoretical research on the loss surface of neural networks can be divided into two main categories: one that studies the global optimality of local minima, see e.g. [7, 13, 14, 17, 18, 19, 21, 22, 25, 27, 28, 29, 30, 34, 38, 42, 45, 46, 47, 48, 50, 51], and the other that studies landscape connectivity. This work falls into the latter category. Specifically, the works of [12, 37] suggest that the loss function of a two-layer network becomes more and more connected as the number of neurons increases. In [43] and [32], it is proved that networks with one sufficiently wide layer have no spurious valleys and connected sublevel sets, respectively. All these works require networks with at least one layer of order $N$ neurons, whereas Corollary 4.3 requires widths of order $\sqrt{N}$ neurons. In [5], the authors study the connectivity of *equivalent* minima (i.e. representing the same network function) which arise by permuting neurons from the same layer. A similar property is investigated in [20] for all local minima lying in a cell or activation region (i.e. those sharing the same activation patterns across all the hidden neurons and training samples). In contrast, here we consider a more general setting: Theorem 4.1 provides upper bounds on the loss when connecting two *arbitrary* points in parameter space. In particular, these points need not be local minima, nor share the same activation patterns, or even represent the same function. This allows us to study the mode connectivity phenomenon of SGD solutions [24], which are typically not captured by the previous assumptions. Other works discuss lower bounds on over-parameterization to ensure the existence of descent paths in shallow nets [39], the satisfiability of the dropout stability condition in the mean field regime [40].

Memorization capacity is an old and well-studied topic [4, 8]. Networks with $\Omega(N)$ parameters are known to memorize $N$ arbitrary samples, and this type of results has been studied for a variety of settings, see [3, 6, 16, 31, 44, 49]. Nevertheless, how memorization capacity is related to the mode connectivity phenomenon remains unclear, especially in the absence of a layer of width $\Omega(N)$. Our work draws a formal connection between the two, which opens up new directions for future works.

## 3 Preliminaries

Given two nonnegative integers $i, j$, let $[i, j] = \{i, \ldots, j\}$ and $[i] = \{1, 2, \ldots, i\}$. Given a set $\mathcal{I}$, we denote by $|\mathcal{I}|$ its cardinality. We consider an $L$-layer neural network with weight matrices $W_l \in \mathbb{R}^{n_{l-1} \times n_l}$ for $l \in [L]$, and bias vectors $b_l \in \mathbb{R}^{n_l}$ for $l \in [L-1]$. Here, $n_0$ is the input dimension, $n_L$ the output dimension, and $n_1, \ldots, n_{L-1}$ the widths of all the hidden layers. Let $\theta$ be the set of all the network parameters. Given an input $x \in \mathbb{R}^{n_0}$, the feature vector $f_l \in \mathbb{R}^{n_l}$ at layer $l$ is defined as

$$f_l(\theta, x) = \begin{cases} x, & l = 0, \\ \sigma(W_l^T f_{l-1}(\theta, x) + b_l), & l \in [L-1], \\ W_L^T f_{L-1}(\theta, x), & l = L, \end{cases} \tag{1}$$

where $\sigma : \mathbb{R} \to \mathbb{R}$ is some activation function. Let $\Omega$ be the parameter space of the network. For a data distribution $(x, y) \sim \mathcal{D}$, the population loss $\Phi : \Omega \to \mathbb{R}$ is defined as

$$\Phi(\theta) = \mathbb{E}_{\mathcal{D}}[\Psi(f_L(\theta, x), y)], \tag{2}$$

where $\Psi(\hat{y}, y)$ is assumed to be convex with respect to the first argument. Typical examples of $\Psi$ which satisfy this assumption include the standard square loss and cross-entropy loss. By letting $\mathcal{D}$ be the uniform distribution over a fixed set of $N$ samples, the loss $\Phi$ reduces to the empirical loss.

In the following, we frequently encounter situations where we are given a layer $l \in [0, L-1]$, and we want to study a subnetwork from layer $l$ to the output layer $L$. The widths and the inputs to this subnet are defined as follows. Let $\mathcal{I}_p \subseteq [n_p]$ be a subset of neurons at layer $p$, for $p \in [l, L-1]$. By convention, let $\mathcal{I}_0 = [n_0]$. Let $f_{p, \mathcal{I}_p} \in \mathbb{R}^{|\mathcal{I}_p|}$ be obtained from $f_p$ by taking the neurons indexed by $\mathcal{I}_p$. The layer widths of this subnet (associated to $l$) are given by $m_l^{(l)} = |\mathcal{I}_l|$, $m_p^{(l)} = n_p - |\mathcal{I}_p|$ for $p \in [l+1, L-1]$, and $m_L^{(l)} = n_L$. The associated weight matrices and bias vectors are $\{U_p^{(l)}\}_{p=l+1}^L$ and $\{v_p^{(l)}\}_{p=l+1}^{L-1}$, where $U_p^{(l)} \in \mathbb{R}^{m_{p-1}^{(l)} \times m_p^{(l)}}$ and $v_p^{(l)} \in \mathbb{R}^{m_p^{(l)}}$. Let $\xi_l = ((U_p^{(l)})_{p=l+1}^L, (v_p^{(l)})_{p=l+1}^{L-1})$, see Figure 2 for an illustration. Given an input $z \in \mathbb{R}^{m_l^{(l)}}$ to this subnetwork, the corresponding output, denoted by $h_L(\xi_l, z) \in \mathbb{R}^{n_L}$, is recursively defined for $p \in [l, L]$ as

$$h_p(\xi_l, z) = \begin{cases} z, & p = l, \\ \sigma\left((U_p^{(l)})^T h_{p-1}(\xi_l, z) + v_p^{(l)}\right), & p \in [l+1, L-1], \\ (U_L^{(l)})^T h_{L-1}(\xi_l, z), & p = L. \end{cases} \tag{3}$$

## 4 Main Result: Statement and Discussion

We now state our main result, which is proved in Section 5.

**Theorem 4.1** *Fix any $\theta_0, \theta_1 \in \Omega$ and any $\epsilon > 0$. Then, there exists a continuous piecewise linear (PWL) path $\theta : [0, 1] \to \Omega$ such that $\theta(0) = \theta_0, \theta(1) = \theta_1$, and the following holds:*

$$\max_{t \in [0,1]} \Phi(\theta(t)) \leq \max_{\theta' \in \{\theta_0, \theta_1\}} \max\left(\Phi(\theta'), \min_{\substack{\mathcal{I}_1 \subseteq [n_1], ..., \mathcal{I}_{L-1} \subseteq [n_{L-1}] \\ s.t. |\mathcal{I}_1| \geq \frac{n_1}{2}, ..., |\mathcal{I}_{L-1}| \geq \frac{n_{L-1}}{2}}} \max_{l \in [0, L-1]} \inf_{\xi_l} \mathbb{E}_{\mathcal{D}}[\Psi(h_L(\xi_l, f_{l, \mathcal{I}_l}(\theta', x)), y)] + \epsilon\right) \tag{4}$$

*where $h_L(\xi_l, f_{l, \mathcal{I}_l}(\theta', x))$ is the output of the subnetwork from layer $l$ to layer $L$ as defined in* (3), *fed with input $f_{l, \mathcal{I}_l}(\theta', x)$, and $\xi_l$ contains the parameters of this subnetwork.*

We remark that the second maximum on the RHS of (4) is intended to be taken between the two components in the open bracket (i.e., $\Phi(\theta')$ and the $\min \max \inf$).

An illustration of the subnets $\xi_l$'s appearing in the theorem can be found in Figure 2. In words, for layer $l \in [L-1]$, we pick a subset $\mathcal{I}_l$ containing at least half the hidden neurons in that layer. Then, the loss over the PWL path is upper bounded by the loss at $L$ points that are formed by connecting the subset of features (indexed by $\mathcal{I}_l$) at layer $l$ to the output via the subnetwork defined in (3). The key idea is that we can construct low-loss paths connecting these sparse points, and the subnetworks guarantee a sufficient amount of sparsity: they have $|\mathcal{I}_l|$ neurons at layer $l$, and $n_p - |\mathcal{I}_p|$ neurons at layer $p \in [l+1, L-1]$. The additive factor $\epsilon$ in the RHS of (4) serves to deal with the case in which the $\inf$ is not attained (as for the cross-entropy loss). We note that Theorem 4.1 also holds for deep nets without biases, which can be proved by following the same argument as in Section 5.

**Equivalent formulation.** Before discussing the implications of Theorem 4.1, we provide an equivalent way to write the "$\min \max \inf$" on the RHS of (4). This characterization will be useful when evaluating numerically our bound in Section 6. First, note that the subnetwork $\xi_l$ in (4) is fully determined by $\mathcal{I}_l$ and the cardinalities of $\mathcal{I}_{l+1}, \ldots, \mathcal{I}_{L-1}$. To this end, let $K_l = |\mathcal{I}_l|, l \in [L-1]$, and

$$Q_l = \inf_{\xi_l} \mathbb{E}_{\mathcal{D}}[\Psi(h_L(\xi_l, f_{l, \mathcal{I}_l}(\theta', x)), y)], \quad l \in [0, L-1]. \tag{5}$$

Then, clearly $Q_l$ is fully determined by the variables $\{\theta', \mathcal{I}_l, K_{l+1}, \ldots, K_{L-1}\}$, and thus it does not depend on the actual content of $\mathcal{I}_k$'s for $k \neq l$. Based on this observation, one can easily argue that

$$\min_{\substack{\mathcal{I}_1 \subseteq [n_1], \ldots, \mathcal{I}_{L-1} \subseteq [n_{L-1}] \\ \text{s.t. } |\mathcal{I}_1| \geq \frac{n_1}{2}, \ldots, |\mathcal{I}_{L-1}| \geq \frac{n_{L-1}}{2}}} \max_{l \in [0, L-1]} Q_l = \min_{\frac{n_1}{2} \leq K_1 \leq n_1, \ldots, \frac{n_{L-1}}{2} \leq K_{L-1} \leq n_{L-1}} \max_{l \in [0, L-1]} \min_{\substack{\mathcal{I}_l \subseteq [n_l] \\ |\mathcal{I}_l| = K_l}} Q_l. \tag{6}$$

To see why (6) holds, we first write the minimum over the sets $\{\mathcal{I}_l\}_{l \in [L-1]}$ on the LHS as a minimum over their cardinalities $\{K_l\}_{l \in [L-1]}$ followed by a minimum over all the possible choices of subsets with fixed cardinalities. From the above discussion, once $K_{l+1}, \ldots, K_{L-1}$ are fixed, the quantity $Q_l$ is entirely independent of $\mathcal{I}_k$ for all $k \neq l$. Thus, one can swap the maximum over $l \in [0, L-1]$ with the minimum over the sets, and the result readily follows.

**A condition milder than dropout stability.** Theorem 1 of [24] can be deduced as a corollary of Theorem 4.1 stated for networks without biases. Let us first recall the definition of $\varepsilon$-dropout stability from [24]. We say that some point $\theta \in \Omega$ is $\varepsilon$-dropout stable if, for all $l \in [L-1]$, there exists a subset of *at most* $\lfloor n_p/2 \rfloor$ hidden neurons in each of the layers $p \in [l, L-1]$ with the following property: after rescaling the outputs of these hidden units by some factor $r$ and setting the outputs of the remaining units to zero, we obtain a parameter $\theta_{l,\mathrm{d}}$ such that $\Phi(\theta_{l,\mathrm{d}}) \leq \Phi(\theta) + \varepsilon$. The proof of the following corollary is postponed to Appendix A.1.

**Corollary 4.2** *Fix any $\theta_0, \theta_1 \in \Omega$ and any $\epsilon > 0$. Assume that $\theta_0$ and $\theta_1$ are $\varepsilon$-dropout stable. Then, there exists a continuous PWL path $\theta : [0, 1] \to \Omega$ such that $\theta(0) = \theta_0, \theta(1) = \theta_1$, and it holds:*

$$\max_{t \in [0,1]} \Phi(\theta(t)) \leq \max\left(\Phi(\theta_0), \; \Phi(\theta_1)\right) + \epsilon + \varepsilon. \tag{7}$$

We highlight that requiring dropout stability to ensure the connectivity of solutions is a much stronger assumption than what is needed in Theorem 4.1. In fact, in order to have that the loss along the path connecting $\theta_0$ and $\theta_1$ is $\varepsilon$-close to the loss at the extremes, Theorem 1 of [24] requires that $\theta_0$ and $\theta_1$ are $\varepsilon$-dropout stable. On the contrary, our main result requires that, for $\theta' \in \{\theta_0, \theta_1\}$,

$$\inf_{\xi_l} \mathbb{E}_{\mathcal{D}}[\Psi\left(h_L\left(\xi_l, f_{l,\mathcal{I}_l}(\theta', x)\right), y\right)] \tag{8}$$

is no bigger than $\Phi(\theta')$. The key difference with respect to [24] is that in (8) we are not restricted to rescaling the weights of the original network (as in the dropout stability assumption), but we are allowed to optimize the weights of the subnetwork. This makes a significant difference in practical settings. In Section 6, we provide examples of networks trained on MNIST and CIFAR-10 such that dropout stability is not satisfied, but our mild condition still holds. Thus, even if the path provided by [24] leads to a large loss, we are still able to connect the SGD solutions via a low-loss path.

**Connections with memorization capacity.** The fact that we can choose arbitrary weights to approach the $\inf$ in (8) provides a link between the connectivity of solutions and the memorization capacity of neural nets. Consider a setting in which there are $N$ data samples $\{(x_j, y_j)\}_{j \in [N]}$ and we wish to minimize the empirical training loss, i.e. $\mathcal{D}$ is the uniform distribution over these samples. By exploiting recent progress on the finite-sample expressivity of neural networks [6, 49], one deduces the corollary below, whose proof is deferred to Appendix A.2. We recall that a set of points $\{x_i\}_{i \in [N]}$ in $\mathbb{R}^d$ is said to be in generic position if any hyperplane in $\mathbb{R}^d$ contains at most $d$ points.

**Corollary 4.3** *Fix any $\theta_0, \theta_1 \in \Omega$ and any $\epsilon > 0$, and let $\sigma$ be ReLU. Let $\{(x_j, y_j)\}_{j \in [N]}$ be $N$ data samples, where $y_j \in [-1, 1]^{n_L}$ for $j \in [N]$ and the vectors $\{x_j\}_{j \in [N]}$ are distinct. Let $\mathcal{D}$ be the uniform distribution over these samples. For $i \in \{0, 1\}$, assume that there exist subsets of neurons $\{\mathcal{I}_l^{(i)}\}_{l \in [L-1]}$ such that the following holds:*

*(A1)* $|\mathcal{I}_{L-1}^{(i)}| \geq \frac{n_{L-1}}{2}$ *and* $4 \left\lfloor \frac{n_{L-2}}{8} \right\rfloor \left\lfloor \frac{n_{L-1} - |\mathcal{I}_{L-1}^{(i)}|}{4n_L} \right\rfloor \geq N.$

*(A1-b) For some $\varepsilon \geq 0$,*

$$\inf_{W \in \mathbb{R}^{|\mathcal{I}_{L-1}^{(i)}| \times n_L}} \mathbb{E}_{\mathcal{D}}[\Psi(W^T f_{L-1, \mathcal{I}_{L-1}^{(i)}}(\theta_i, x), y)] \leq \Phi(\theta_i) + \varepsilon. \tag{9}$$

*(A2)* $|\mathcal{I}_{L-2}^{(i)}| = \lceil \frac{n_{L-2}}{2} \rceil$, *and the feature vectors* $\{f_{L-2, \mathcal{I}_{L-2}^{(i)}}(\theta_i, x_j)\}_{j \in [N]}$ *are in generic position.*

*(A3) For all $l \in [L-3]$: $|\mathcal{I}_l^{(i)}| = n_l - 1$, and the feature vectors $\{f_{l,\mathcal{I}_l^{(i)}}(\theta_i, x_j)\}_{j \in [N]}$ are distinct.*

*Then, there exists a continuous PWL path $\theta : [0,1] \to \Omega$ s.t. $\theta(0) = \theta_0, \theta(1) = \theta_1$, and it holds:*

$$\max_{t \in [0,1]} \Phi(\theta(t)) \leq \max\left(\Phi(\theta_0),\ \Phi(\theta_1)\right) + \epsilon + \varepsilon. \tag{10}$$

This result should be regarded as a proof of concept. Improved bounds on memorization capacity[1] would immediately lead to stronger results about connectivity. Let us now discuss in more detail the assumptions needed in Corollary 4.3. Assumption (A2) ensures sufficient "diversity" among the features at the second last hidden layer: the vectors $\{f_{L-2,\mathcal{I}_{L-2}^{(i)}}(\theta_i, x_j)\}_{j \in [N]}$ need to be in generic position, which is rather standard in the literature about the memorization capacity of neural networks, see e.g. [4]. Assumption (A3) requires a minimal condition that the feature vectors $\{f_{l,\mathcal{I}_l^{(i)}}(\theta_i, x_j)\}_{j \in [N]}$ are distinct for $l \in [L-3]$. Assumption (A1-b) involves the quality of the features at the last hidden layer: it requires that, by removing $n_{L-1} - |\mathcal{I}_{L-1}^{(i)}|$ neurons and picking suitable new outcoming weights for the remaining ones, we do not increase the loss by more than $\varepsilon$. Assumption (A1) involves the over-parameterization of the last two hidden layers of the network: it requires that $n_{L-2}(n_{L-1} - |\mathcal{I}_{L-1}^{(i)}|)$ is at least roughly $8Nn_L$. Pick $|\mathcal{I}_{L-1}^{(i)}| = (1-\delta)n_{L-1}$, and assume that $n_L$ is a constant independent of $N$. Then, assumption (A1-b) holds if the loss remains stable after removing a $\delta$-fraction of the neurons, and assumption (A1) holds if $n_{L-2}n_{L-1} = \Omega(N)$. We remark that there is no requirement on the widths of the remaining layers.

It is shown in [33] that, for deep nets, if the first hidden layer has $N+1$ neurons then one gets connected sublevel sets. This bound is tight as a 2-layer network of width $N$ can have disconnected sublevel sets. Here, we use properties of the features of the solutions we wish to connect in order to relax the requirement on the over-parameterization. In particular, if the features at the last two hidden layers are good enough (i.e. removing a small number of neurons at layer $L-1$ does not increase the loss much, and a subset of features at layer $L-2$ are in generic position), then $\Omega(\sqrt{N})$ neurons in each of these two layers are sufficient to ensure the connectivity. More generally, the fact that we are free to choose the sizes of the sets $\mathcal{I}_1, \ldots, \mathcal{I}_{L-1}$ in the RHS of (4) allows one to trade-off between the quality of the features at a given layer and the amount of over-parameterization in the layers above it.

**Linearly separable features.** One extreme example of this trade-off is when the features are linearly separable (i.e. the features are highly representative of the labels). More formally, consider a classification task where one is given the labels $y_j \in \{0,1\}^{n_L}$, and the data samples are linearly separable. Recall that, a set of data samples $\{(x_j, y_j)\}_{j \in [N]}$ is called "linearly separable" if there exist $n_L$ hyperplanes, each separating the samples of one class from the other classes perfectly. Pick two points such that, by removing $n_L$ neurons per layer, their features remain linearly separable at each layer. Then, these points can be connected by a low-loss path (without any extra over-parameterization requirement). This is formalized in the following corollary, whose proof appears in Appendix A.3.

**Corollary 4.4** *Fix any $\theta_0, \theta_1 \in \Omega$ and any $\epsilon > 0$. Assume that $n_l \geq 2n_L$ for $l \in [L-1]$ and that there exists an interval $[\alpha, \beta]$ s.t. $\sigma$ is strictly monotonic on $[\alpha, \beta]$. Let $\{(x_j, y_j)\}_{j \in [N]}$ be $N$ linearly separable data samples, where $y_j \in \{0,1\}^{n_L}$ for $j \in [N]$. Let $\mathcal{D}$ be the uniform distribution over these samples, and let $\Psi$ be the standard cross-entropy loss. Assume that, for $i \in \{0,1\}$, there exists subsets $\{\mathcal{I}_l^{(i)}\}_{l \in [L-1]}$ such that $\frac{n_l}{2} \leq |\mathcal{I}_l^{(i)}| \leq n_l - n_L$ and the set of features $\{f_{l,\mathcal{I}_l^{(i)}}(\theta_i, x_j), y_j\}_{j \in [N]}$ is linearly separable, for all $l \in [0, L-1]$. Then, there exists a continuous PWL path $\theta : [0,1] \to \Omega$ such that $\theta(0) = \theta_0, \theta(1) = \theta_1$, and the following holds:*

$$\max_{t \in [0,1]} \Phi(\theta(t)) \leq \max\left(\Phi(\theta_0),\ \Phi(\theta_1)\right) + \epsilon. \tag{11}$$

We remark that the result of Corollary 4.4 is not restricted to the cross-entropy loss, but it holds for any loss function $\Psi$ whose infimum is 0 when the input is linearly separable. Furthermore, the assumption on the activation function being monotonic in an interval is very mild, and it serves to avoid degenerate cases (e.g. constant activation function).

---

[1]E.g., less parameters in the interpolating network or less assumptions on the input data.

# 5 Proof of Theorem 4.1

In the following, the "incoming weights" of a neuron refer to both the incoming weight vector and the bias of that neuron, and the "outcoming weights" of a neuron simply refer to its outcoming weight vector. We call a neuron "inactive" if its incoming weights are zero, and we call a neuron "pure zero" if both its incoming and outcoming weights are zero. Clearly, a pure zero neuron is inactive. Whenever we say "set some parameter $w$ from $a$ to $b$", we simply mean to change the current value of $w$ from $a$ to $b$ by following a direct line segment: $w(\lambda) = (1 - \lambda)a + \lambda b$, where $\lambda \in [0, 1]$. First, we state and prove a useful claim. Then, we present the proof of Theorem 4.1.

**Claim 5.1** *Consider two distinct hidden neurons $p$ and $q$ on the same layer, and assume that $p$ is a pure zero neuron. Then, one can swap $p$ and $q$ by a continuous PWL path along which the network output is invariant.*

**Proof of Claim 5.1:** Let $p_{\text{in}}$ and $p_{\text{out}}$ denote the incoming resp. outcoming weights of $p$. Similarly, we denote by $q_{\text{in}}$ and $q_{\text{out}}$ the incoming and outcoming weights of $q$. We change $p$ and $q$ continuously by applying the following paths sequentially (after each step, we reset the value of $p$ and $q$ to the ending point of the path): *(i)* $p_{\text{in}}(\lambda) = (1 - \lambda)p_{\text{in}} + \lambda q_{\text{in}}$, *(ii)* $[p_{\text{out}}(\lambda), q_{\text{out}}(\lambda)] = [\lambda q_{\text{out}}, (1 - \lambda)q_{\text{out}}]$, and *(iii)* $q_{\text{in}}(\lambda) = (1 - \lambda)q_{\text{in}}$. For all these paths, $\lambda$ goes from 0 to 1. Note that path *(i)* does not change the network output, since the outcoming weights of $p$ are zero. At the end of the first path, $p$ and $q$ have the same incoming weights (i.e. they are identical neurons). Furthermore, path *(ii)* does not change the network output, since the total contribution of $p$ and $q$ to the next layer is constant along the path. Finally, also path *(iii)* does not change the network output, since the outcoming weights of $q$ are zero at the end of path *(ii)*. After performing these three steps, $p$ and $q$ are swapped, i.e. $q$ becomes pure zero while $p$ has the original weights of $q$. □

**Proof of Theorem 4.1:** Fix the subsets $\{\mathcal{I}_l\}_{l \in [L-1]}$, where $\mathcal{I}_l \subseteq [n_l]$, $|\mathcal{I}_l| \geq n_l/2$, and let $\bar{\mathcal{I}}_l$ be the complement of $\mathcal{I}_l$. We show that there is a continuous PWL path from $\theta_0$ to some other point for which all the neurons in $\{\bar{\mathcal{I}}_l\}_{l \in [L-1]}$ are pure zero, and the loss along the path is at most

$$\max\left(\Phi(\theta_0), \max_{l \in [0, L-1]} \inf_{\xi_l} \mathbb{E}_{\mathcal{D}}[\Psi\left(h_L\left(\xi_l, f_{l,\mathcal{I}_l}(\theta_0, x)\right), y\right)] + \epsilon\right). \tag{12}$$

To prove the claim, we start from $\theta_0$ and apply a sparsification procedure that makes the neurons in $\bar{\mathcal{I}}_{L-1}, \ldots, \bar{\mathcal{I}}_1$ become pure zero, one layer at a time from layer $L - 1$ down to layer 1. First, we construct a path that makes the outcoming weights of the neurons in $\bar{\mathcal{I}}_{L-1}$ become zero by changing only the weights in the last layer. Note that $\Psi$ is convex in the weights of the last layer, hence $\Phi$ is also convex in those weights. Therefore, the loss along this path is at most

$$\max\left(\Phi(\theta_0), \inf_{\xi_{L-1}} \mathbb{E}_{\mathcal{D}}[\Psi\left(h_L\left(\xi_{L-1}, f_{L-1,\mathcal{I}_{L-1}}(\theta_0, x)\right), y\right)] + \epsilon\right).$$

Once the outcoming weights of a neuron are all zero, the incoming weights can also be set to zero without changing the loss. Thus, the neurons in $\bar{\mathcal{I}}_{L-1}$ can be made pure zero.

Now, we assume via induction that we have already made all the neurons in $\{\bar{\mathcal{I}}_{l+1}, \ldots, \bar{\mathcal{I}}_{L-1}\}$ pure zero, and we want to make the neurons in $\bar{\mathcal{I}}_l$ pure zero. It suffices to make all the inter-connections between $\bar{\mathcal{I}}_l$ and $\mathcal{I}_{l+1}$ become zero, because the connections between $\bar{\mathcal{I}}_l$ and $\bar{\mathcal{I}}_{l+1}$ are already zero due to the inactivity of $\bar{\mathcal{I}}_{l+1}$. To do this, one first observes that the pure-zero state of the neurons in $\{\bar{\mathcal{I}}_{l+1}, \ldots, \bar{\mathcal{I}}_{L-1}\}$ separates the original network from layer $l + 1$ to layer $L - 1$ into two parallel subnetworks: one with neurons in $\{\mathcal{I}_p\}_{p=l+1}^{L-1}$, and the other with pure zero neurons. Now, set the incoming weights of the neurons in $\{\bar{\mathcal{I}}_{l+1}, \ldots, \bar{\mathcal{I}}_{L-1}\}$ to the corresponding values given by the subnetwork $\xi_*$, where $\xi_*$ satisfies the following

$$\mathbb{E}_{\mathcal{D}}[\Psi\left(h_L\left(\xi_*, f_{l,\mathcal{I}_l}(\theta_0, x)\right), y\right)] \leq \inf_{\xi_l} \mathbb{E}_{\mathcal{D}}[\Psi\left(h_L\left(\xi_l, f_{l,\mathcal{I}_l}(\theta_0, x)\right), y\right)] + \epsilon. \tag{13}$$

Note that a subnetwork $\xi_*$ which satisfies (13) always exists: if the infimum on the RHS of (13) is actually a minimum, then we can pick $\xi_*$ to be one of the minimizers. Otherwise, there exists a $\xi_*$ whose loss approaches the infimum with arbitrarily small error. In the above step, only the incoming weights that are relevant to the subnetwork $\xi_*$ are reset, while the other ones are kept unchanged. In particular, this step does not affect the outputs of the neurons in the other subsets $\{\mathcal{I}_{l+1}, \ldots, \mathcal{I}_{L-1}\}$ as well as the output of the network. Next, let us set all the outcoming weights

in $\bar{\mathcal{I}}_{L-1}$ to the corresponding value given by $\xi_*$, and all the outcoming weights in $\mathcal{I}_{L-1}$ to zero (in one step). By the convexity of $\Psi$ w.r.t. the output weights, we have that the loss along this path is upper bounded by the loss at the extremes of the path, which is in turn upper bounded by (12). Now, since the outcoming weights of $\mathcal{I}_{L-1}$ are zero, it follows that the parallel subnetwork associated to $\{\mathcal{I}_{l+1}, \ldots, \mathcal{I}_{L-1}\}$ does not contribute to the network output. Thus, we can change the incoming weights of all the neurons in this subnetwork to any arbitrary values without affecting the loss. Hence, we turn all these neurons into inactive and then pure zero, without changing the loss. As a result, all the inter-connections between $\bar{\mathcal{I}}_l$ and $\mathcal{I}_{l+1}$ will become zero, as wanted from above. At this point, note that the weights of the neurons in $\{\bar{\mathcal{I}}_{l+1}, \ldots, \bar{\mathcal{I}}_{L-1}\}$ are those of $\xi_*$. Thus, in order to finish the induction proof, we need to make them become pure zero again. This can be done by swapping, at each layer $p \in [l+1, L-1]$, one pure zero neuron in $\mathcal{I}_p$ with one neuron in $\bar{\mathcal{I}}_p$. As $|\mathcal{I}_p| \geq \frac{n_p}{2} \geq |\bar{\mathcal{I}}_p|$, this is always possible, and the network output is invariant by Claim 5.1.

The sparsification procedure described above can be performed for any subsets $\{\mathcal{I}_l\}_{l \in [L-1]}$ s.t. $|\mathcal{I}_l| \geq \frac{n_l}{2}$ for $l \in [L-1]$. Thus, we can take the minimum in (12) over such subsets. Let $\{\mathcal{I}_l^*\}_{l \in [L-1]}$ be the optimal subsets for $\theta_0$. Let us apply the same sparsification procedure to $\theta_1$. We can assume w.l.o.g. that the optimal subsets for $\theta_1$ and $\theta_0$ are the same. In fact, if this is not the case, we use Claim 5.1 and swap the pure zero neurons with other ones on the same layer in order to satisfy this property. At this point, both $\theta_0$ and $\theta_1$ have been sparsified, and they have the same set of pure zero neurons at each layer, indexed by $\{\bar{\mathcal{I}}_l^*\}_{l \in [L-1]}$.

Finally, we connect the sparsified versions of $\theta_0$ and $\theta_1$. Let $A$ be the subnetwork consisting of all the hidden neurons in $\{\bar{\mathcal{I}}_l^*\}_{l \in [L-1]}$, and $B$ the subnetwork consisting of all the remaining neurons. Both $A$ and $B$ share the same input and output layers as the original network. Note also that the hidden layers of $B$ have larger widths than the corresponding layers of $A$. Since $A$ and $B$ are decoupled, one can turn one on and the other off by setting the appropriate output weights to zero, and then update the incoming weights of the other subnetwork to any desired value without affecting the loss. In this way, one can make all the incoming weights (restricted to each parallel subnetwork) of $\theta_0$ equal to the corresponding values of $\theta_1$. Finally, one uses the convexity of the loss $\Psi$ again to equalize the output weights of both $\theta_0$ and $\theta_1$. Hence, the loss incurred to connect the sparsified versions of $\theta_0$ and $\theta_1$ is no bigger than

$$\max_{\theta' \in \{\theta_0, \theta_1\}} \inf_{\xi_0} \mathbb{E}_{\mathcal{D}}[\Psi(h_L(\xi_0, f_0(\theta', x)), y)] + \epsilon.$$

By taking the maximum of the loss along the paths considered in this proof, we get the bound (4). $\square$

# 6 Numerical Experiments

We compare the losses along the path of Theorem 4.1 and the one in [24] which our theoretical analysis has improved upon. Implementation details and a link to the code are given in Appendix C.

**Datasets and architectures.** We consider MNIST [26] and CIFAR-10 [23] datasets. As a preprocessing stage, we perform a scaling into $[-1, 1]$ for MNIST, and data whitening for CIFAR-10. For MNIST, we consider a fully connected network (FCN) with 10 hidden layers ($L = 11$), each of width 245. For CIFAR-10, we consider two architectures: *(i)* an FCN with 5 hidden layers ($L = 6$), each of width 500, and *(ii)* the VGG-11 network [41]. All architectures have ReLU activations. We train each network by standard SGD with cross-entropy loss, batch size 100 and no explicit regularizers. All the networks are trained beyond the point where the dataset is perfectly separated, and no early stopping mechanism is used. For each trained network, we perform the following two experiments.

**Experiment (A):** we evaluate the RHS of (4), with $\theta'$ being the trained model, by using the equivalent characterization (6). First, we fix the integers $K_l = \lceil n_l/2 \rceil$. Then, for each $l \in [0, L-1]$, we compute the quantity $\tilde{Q}_l$, defined as:

$$\tilde{Q}_l := \min_{\substack{\mathcal{I}_l \subseteq [n_l] \\ |\mathcal{I}_l| = K_l}} Q_l = \min_{\substack{\mathcal{I}_l \subseteq [n_l] \\ |\mathcal{I}_l| = K_l}} \inf_{\xi_l} \mathbb{E}_{\mathcal{D}}[\Psi(h_L(\xi_l, f_{l, \mathcal{I}_l}(\theta', x)), y)]. \tag{14}$$

We upper bound the inf in (14) by running an independent SGD routine, and upper bound the min there by sampling the subsets $\mathcal{I}_l$ uniformly at random for $T_A = 20$ times and by picking the best training loss resulting from the inf. The resulting bounds on $\tilde{Q}_l$ are shown by the blue curves in

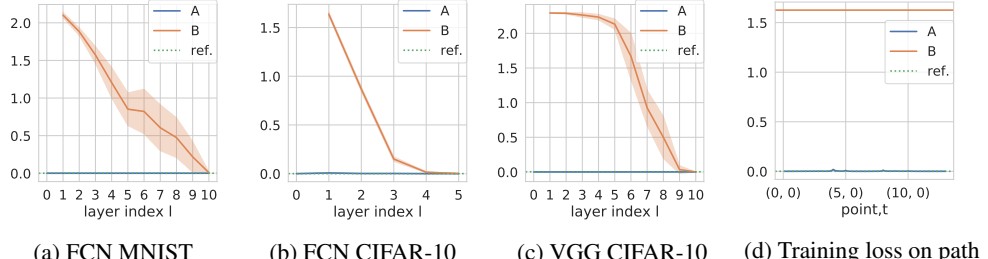

| (a) FCN MNIST | (b) FCN CIFAR-10 | (c) VGG CIFAR-10 | (d) Training loss on path |

Figure 3: (a)–(c): Training loss for Experiments (A) and (B). The dotted green line is the originally trained model $\theta'$, used as a reference. Mean and 95% confidence interval are shown for 10 different trained networks $\theta'$ in (a)-(b), and for 5 different $\theta'$ in (c). The plot (d) shows the loss along the paths of Experiments (A) and (B) between a fixed pair of solutions for the FCN trained on CIFAR-10.

Figure 3. Note that by taking the maximum of these values for $l \in [0, L-1]$, one obtains an upper bound on (6), which is exactly the $\min\max\inf$ term in Theorem 4.1. As Theorem 4.1 requires at least two different points $\theta'$ to connect, in Figure 3 we run our experiments multiple times (i.e. by training multiple network $\theta'$ above), and we report the mean together with the 95% confidence interval. The performance of these original models $\theta'$'s is shown by the horizontal dotted green line in Figure 3 (used as a reference). As one can see, the blue curve is always close to the reference curve, which means that our upper bound as given in Theorem 4.1 for connecting these $\theta'$ solutions is always close to their original training losses, and thus all these solutions are approximately connected.

**Experiment (B):** we test the dropout stability of the trained model as defined in [24]. For each $l \in [L-1]$, we proceed as follows: *(i)* randomly remove half of the neurons at all layers $p \in [l, L-1]$, and *(ii)* optimize the scaling factor of the output of the remaining network by minimizing the training loss. This last optimization is performed in our experiments via the Scipy method `minimize_scalar(method="brent")`. For each $l \in [L-1]$, this procedure (i.e. do *(i)*, and then *(ii)*) is repeated $T_B = 200$ times, and we pick the best result based on the training loss. We have chosen the value $T_B = 200$ by observing that, if we increase $T_B$ from 100 to 200, the change in the minimum loss is negligible. The results here are shown by the orange curves in Figure 3. One observes that dropout stability tends to be satisfied at the layers near the output of the network, while dropping the neurons from the first few layers increases the loss significantly. This is consistent with the findings of [24], where it is reported that dropout stability does not hold for the first few layers of the VGG-11 architecture without dropout training. If one instead uses dropout during training, [24] show that the trained model satisfies the dropout stability property. In Experiment (A), we have shown that, even when dropout is *not* used as an explicit regularizer during training (and consequently dropout stability may not hold), we can still ensure the existence of low-loss connecting paths. This is because our bound in Theorem 4.1 allows the optimization of the weights of the subnetwork $\xi_l$.

Figure 3(d) illustrates the paths between two SGD solutions obtained from Experiment (A) and (B) for the FCN with $L = 6$ trained on CIFAR-10. While the path (A) constructed via Theorem 4.1 has loss close to zero (and close to the loss of the reference model), the path of [24] experiences a large loss. Additional results in Appendix B show similar patterns for the test error (see Figure 6) and for different dropout ratios (see Figures 7-8). Furthermore, the log-scale plot in Figure 5 of Appendix B allows one to zoom into the detailed difference between our bound and the reference.

In Figure 4, we perform Experiments (A)-(B) for an FCN with 2 hidden layers of equal width, trained on MNIST and CIFAR-10. Here, we vary the width, and report the maximum of the bounds over all the layers. As expected, the loss gets smaller as width increases. However, the bound provided by Theorem 4.1 is close to 0 already for networks of moderate sizes: about 200 neurons for MNIST, and 500 for CIFAR-10. These networks widths roughly correspond to $\sqrt{N}$, as predicted by Corollary 4.3. On the contrary, much larger widths are necessary for dropout stability to hold: 2000 neurons for MNIST, and for CIFAR-10, networks with 3000 neurons are still quite far from being dropout stable.

Taking all these experimental results together, we have demonstrated that, for simple datasets (e.g. MNIST) and for large enough widths, dropout stability holds. However, for more difficult learning tasks (e.g. CIFAR-10), and for networks of moderate widths or larger depths, there is a strong

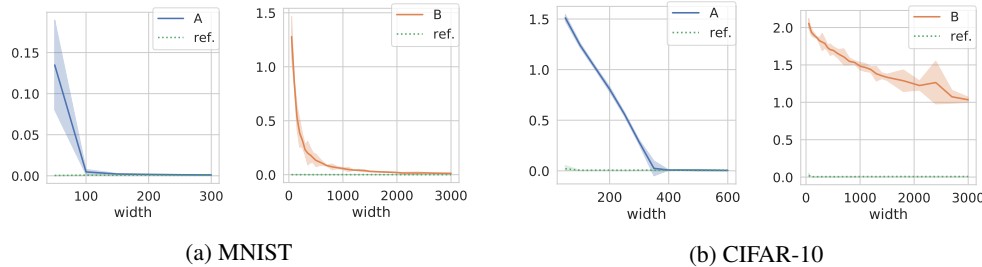

|   (a) MNIST   |   (b) CIFAR-10   |

Figure 4: Training loss (maximum over all layers) for Experiments (A) and (B) for an FCN with two hidden layers of equal albeit varying widths. The dotted line is the originally trained model. Mean and 95% confidence interval are shown for 3 different solutions $\theta'$.

evidence that dropout stability is not satisfied. Yet, our main theorem provides connecting paths with low loss, even when the over-parameterization of the trained networks is moderate.

## 7  Concluding Remarks

We provide a new characterization for SGD solutions (or more generally, for any two points in parameter space) to be connected by low-loss paths. Numerical experiments demonstrate that the proposed sufficient condition is satisfied by trained networks, also in settings where the requirements of the previous works do not hold. Our results and analysis lead directly to the following open question: *Is having (more than two) hidden layers with $\Omega(\sqrt{N})$ neurons in general sufficient to guarantee the connectivity of all the sublevel sets?* Although width $N$ is both necessary and sufficient for two-layer nets [33], our Corollary 4.3 gives hope that this bound may be not tight for deeper nets, and sublinear widths there might suffice. Answering this question would help us better understand the role of depth on shaping the geometry of neural net loss function. It is also interesting to study: *(i)* Why does SGD converge to solutions fulfilling the trade-off we describe here between feature quality and over-parameterization? *(ii)* Does SGD provably converge to a global optimum for such networks? This last question was studied, e.g. in [1, 10, 35, 36, 52] for regression tasks, but the current results still require at least one layer with $\Omega(N)$ neurons. The analysis here suggests that SGD may explore a well-behaved region of the loss landscape even for networks with sublinear layer widths.

## Acknowledgements

MM was partially supported by the 2019 Lopez-Loreta Prize. QN and PB acknowledge support from the European Research Council (ERC) under the European Union's Horizon 2020 research and innovation programme (grant agreement no 757983).

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
