# Supplementary Material (Appendix)

# When Are Solutions Connected in Deep Networks?

## A   Missing Proofs

### A.1   Proof of Corollary 4.2

**Proof:** Let $\theta' \in \{\theta_0, \theta_1\}$. Then, as $\theta'$ is $\varepsilon$-dropout stable, we obtain that, for every $l \in [L-1]$, there exist subsets of indices $\{\mathcal{I}_p^{l,\mathrm{d}}\}_{p=l}^{L-1}$, with $\mathcal{I}_p^{l,\mathrm{d}} \subseteq [n_p]$ and $\left|\mathcal{I}_p^{l,\mathrm{d}}\right| \leq \lfloor n_p/2 \rfloor$, such that the following property holds: after rescaling the outputs of the hidden units indexed by $\{\mathcal{I}_p^{l,\mathrm{d}}\}_{p=l}^{L-1}$ and setting the outputs of the remaining units to zero, we obtain $\theta'_{l,\mathrm{d}}$ satisfying $\Phi(\theta'_{l,\mathrm{d}}) \leq \Phi(\theta') + \varepsilon$. Now, one observes that, by using Theorem 4.1, it suffices to show that there exists a choice of $\{\mathcal{I}_j\}_{j \in [L-1]}$ with $\mathcal{I}_j \subseteq [n_j]$ and $|\mathcal{I}_j| \geq n_j/2$ such that, for $l \in [0, L-1]$, it holds

$$\inf_{\xi_l} \mathbb{E}_{\mathcal{D}}[\Psi(h_L(\xi_l, f_{l,\mathcal{I}_l}(\theta', x)), y)] \leq \Phi(\theta'_{l,\mathrm{d}}), \tag{15}$$

with $\mathcal{I}_0 = [n_0]$.

For $l \in [L-1]$, pick $\mathcal{I}_l$ such that $\mathcal{I}_l \supseteq \mathcal{I}_l^{l,\mathrm{d}}$ and $|\mathcal{I}_l| = \lceil n_l/2 \rceil$. This is possible since $\left|\mathcal{I}_l^{l,\mathrm{d}}\right| \leq \lfloor n_l/2 \rfloor$. Note that since $|\mathcal{I}_l| = \lceil n_l/2 \rceil$, the requirement that $|\mathcal{I}_l| \geq n_l/2$ from (4) is fulfilled. As $\mathcal{I}_l \supseteq \mathcal{I}_l^{l,\mathrm{d}}$, $f_{l,\mathcal{I}_l}(\theta', x)$ contains all the non-zero elements of the feature vector $f_l(\theta'_{l,\mathrm{d}}, x)$ (i.e. after setting the outputs of the remaining neurons to zero as described above). Furthermore, recall that $\xi_l$ contains the parameters of a subnetwork from layer $l$ to $L$ with layer widths $m_l^{(l)} = |\mathcal{I}_l|$, $m_p^{(l)} = n_p - |\mathcal{I}_p| = \lfloor n_p/2 \rfloor$ for $p \in [l+1, L-1]$ and $m_L^{(l)} = n_L$. Thus, since $\theta'_{l,\mathrm{d}}$ contains at most $\lfloor n_p/2 \rfloor$ non-zero neurons at layer $p \in [l+1, L-1]$, it follows that the network computed at $\theta'_{l,\mathrm{d}}$ can be implemented by the subnetwork. In other words, there exists $\xi_l$ such that $h_L(\xi_l, f_{l,\mathcal{I}_l}(\theta', x)) = f_L(\theta'_{l,\mathrm{d}}, x)$, for all $x$. Hence, (15) holds and the desired claim follows.    $\square$

### A.2   Proof of Corollary 4.3

**Proof:** We show that, for $i \in \{0, 1\}$ and for $l \in [0, L-1]$,

$$\inf_{\xi_l} \mathbb{E}_{\mathcal{D}}\left[\Psi\left(h_L\left(\xi_l, f_{l,\mathcal{I}_l^{(i)}}(\theta_i, x)\right), y\right)\right] \leq \Phi(\theta_i) + \varepsilon, \tag{16}$$

with $\mathcal{I}_0^{(i)} = [n_0]$. When $l = L-1$, (16) is equivalent to (9), which holds by assumption (A1-b). When $l = L-2$, we note that $\{f_{L-2,\mathcal{I}_{L-2}^{(i)}}(\theta_i, x_j)\}_{j \in [N]}$ are in generic position by assumption (A2), and assumption (A1) holds. Thus, (16) follows by applying Proposition 4 of [6] to each of the $n_L$ outputs of the network. When $l \in [0, L-3]$, we note that $\{f_{l,\mathcal{I}_l^{(i)}}(\theta_i, x_j)\}_{j \in [N]}$ are distinct by assumption (A3), the samples $\{x_j\}_{j \in [N]}$ are distinct by assumption, and $y_j \in [-1, 1]^{n_L}$ for $j \in [N]$. Thus, by using again assumption (A1), we can apply Corollary A.1 of [49] and (16) readily follows.[2] Finally, one can easily check that $|\mathcal{I}_l^{(i)}| \geq n_l/2$ for all $l \in [L-1], i \in \{0, 1\}$, which satisfies the requirement in (4). Thus, the desired claim follows from Theorem 4.1.    $\square$

---

[2]Note that we apply Corollary A.1 of [49] with $m = 1$ and $l_1 = L-2$. In this setting, $[m-1]$ and $[l_m + 2 : L-1]$ are the empty set, hence the 1st, 3rd and 4th conditions of Corollary A.1 hold. To verify the 2nd condition, note that $r_1 = 0$, $d_{L-2} = n_{L-2} - |\mathcal{I}_{L-2}^{(i)}| = \lfloor \frac{n_{L-2}}{2} \rfloor$, $d_{L-1} = n_{L-1} - |\mathcal{I}_{L-1}^{(i)}|$, and $d_y = n_L$. Thus, the 2nd condition follows from assumption (A1), and the application of Corollary A.1 is justified.

## A.3 Proof of Corollary 4.4

**Proof:** We show that, for $i \in \{0, 1\}$ and for $l \in [0, L-1]$,

$$\inf_{\xi_l} \mathbb{E}_{\mathcal{D}} \left[ \Psi \left( h_L \left( \xi_l, f_{l, \mathcal{I}_l^{(i)}}(\theta_i, x) \right), y \right) \right] = 0, \tag{17}$$

with $\mathcal{I}_0^{(i)} = [n_0]$. Recall that, by hypothesis, $\{f_{l, \mathcal{I}_l^{(i)}}(\theta_i, x_j), y_j\}_{j \in [N]}$ are linearly separable for $l \in [0, L-1]$. For $k \in [n_L]$, denote by $(y_j)_k$ the $k$-th component of the vector $y_j \in \{0, 1\}^{n_L}$.

First, consider the case $l = L - 1$. Then, the problem in (17) is a convex learning problem with linearly separable inputs, namely $\{f_{L-1, \mathcal{I}_{L-1}^{(i)}}(\theta_i, x_j), y_j\}_{j \in [N]}$. Since $\Psi$ is the cross-entropy loss, (17) holds immediately.

Consider now the case $l \in [0, L-2]$. Then, there exists $\gamma \in (\alpha, \beta)$, a set of $n_L$ weights $w_1, \ldots, w_{n_L} \in \mathbb{R}^{|\mathcal{I}_l^{(i)}|}$ and a set of $n_L$ biases $c_1, \ldots, c_{n_L} \in \mathbb{R}$ such that the following holds: if $(y_j)_k = 1$, then $\langle f_{l, \mathcal{I}_l^{(i)}}(\theta_i, x_j), w_k \rangle + c_k \in (\gamma, \beta)$; otherwise, $\langle f_{l, \mathcal{I}_l^{(i)}}(\theta_i, x_j), w_k \rangle + c_k \in (\alpha, \gamma)$. In words, this means that the pre-activation output of neuron $k$ at layer $l + 1$ maps all the samples of class $k$ into $(\gamma, \beta)$ and the remaining samples into $(\alpha, \gamma)$. Let us assume w.l.o.g. that $\sigma$ is strictly monotonically increasing on $(\alpha, \beta)$. Then, we have that, if $(y_j)_k = 1$, $\sigma(\langle f_{l, \mathcal{I}_l^{(i)}}(\theta_i, x_j), w_k \rangle + c_k) \in (\sigma(\gamma), \sigma(\beta))$; otherwise, $\sigma(\langle f_{l, \mathcal{I}_l^{(i)}}(\theta_i, x_j), w_k \rangle + c_k) \in (\sigma(\alpha), \sigma(\gamma))$. Hence, at layer $l + 1$, we have $n_L$ neurons, each perfectly separating the samples of one class from the others. This shows that the set of features formed by these neurons is linearly separable. By repeating this argument, we can choose the weights of $n_L$ neurons from the next layers $l + 2, \ldots, L - 1$ such that this linear separability property is maintained. Consequently, the subnetwork implemented by $\xi_l$ (which has at least $n_L$ neurons at each of its hidden layers) can perfectly separate the features $\{f_{l, \mathcal{I}_l^{(i)}}(\theta_i, x_j), y_j\}_{j \in [N]}$. Thus, (17) holds, and the desired claim follows from Theorem 4.1. $\square$

# B  Additional Numerical Experiments

**Training loss in logarithmic scale.** Figure 5 plots the loss reported in Figure 3 on a logarithmic $y$-axis. This allows one to see more clearly the difference between our bound (A) and the reference model. We note that this difference remains small across all layers of the network and experiments. Likewise, Figure 9 plots the loss reported in Figure 4 on a logarithmic $y$-axis. The logarithmic scale of the plot magnifies the drop in the metric around a certain threshold, about 100 neurons for MNIST and 400 for CIFAR-10.

**Test error.** Figure 6 shows the corresponding test errors for the experiment in Figure 3. One can observe a similar behavior that our bound (A) continues to remain close to the reference model, while the test error (B) coming from the dropout stability assumption is large at the first layers.

**Different dropout ratios.** In Figures 7-8, we perform Experiments (A) and (B) again with other values of $K_l$'s – the cardinalities of the subsets of neurons $\mathcal{I}_l$'s. Intuitively, in our proofs $K_l$ can be seen as the number of neurons that we choose to keep at each layer, and thus it can also be defined via a certain dropout ratio. For simplicity, we fix this ratio to be the same at every layer, say $p \in [0, 1]$. Let $K_l = \lceil (1 - p) n_l \rceil$, for $l \in [L - 1]$. Then, one notes that the results in Figures 3-4 have been obtained by taking $p = \frac{1}{2}$. Now, we repeat these experiments for $p \in \{\frac{1}{3}, \frac{1}{4}\}$, and report the results in Figures 7-8, respectively. As for Experiment (A), this means that, while more neurons are preserved at each hidden layer (due to smaller $p$), the capacities of the subnetworks $\xi_l$'s from Theorem 4.1 are smaller. As for Experiment (B), since more neurons are kept at each layer, there is a better chance that the dropout stability assumption is satisfied. Let us remark that the results of [24] show the existence of a low-loss path for $p \geq \frac{1}{2}$, whereas our Theorem 4.1 also applies to $p \leq \frac{1}{2}$. The plots show that although experiment (B) benefits from a lower $p$, there is still a significant performance gap with the originally trained model, especially at the lower layers. Meanwhile, this gap is consistently smaller for (A) across different values of $p$. This is on par with the case $p = \frac{1}{2}$ as reported in Section 6.

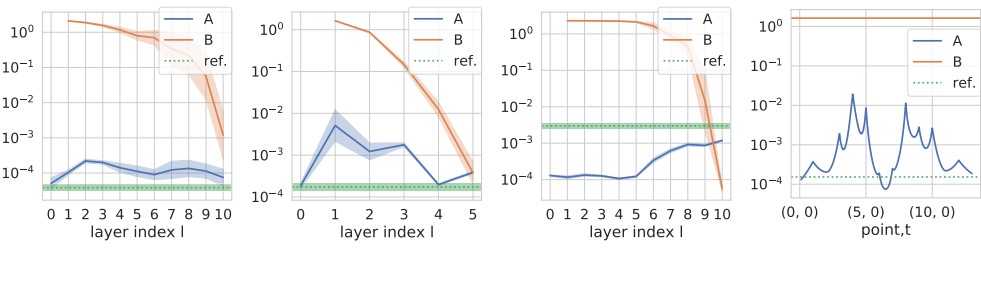

(a) FCN MNIST      (b) FCN CIFAR-10      (c) VGG CIFAR-10      (d) Training loss on path

Figure 5: Results of Figure 3 plotted with a logarithmic $y$-axis.

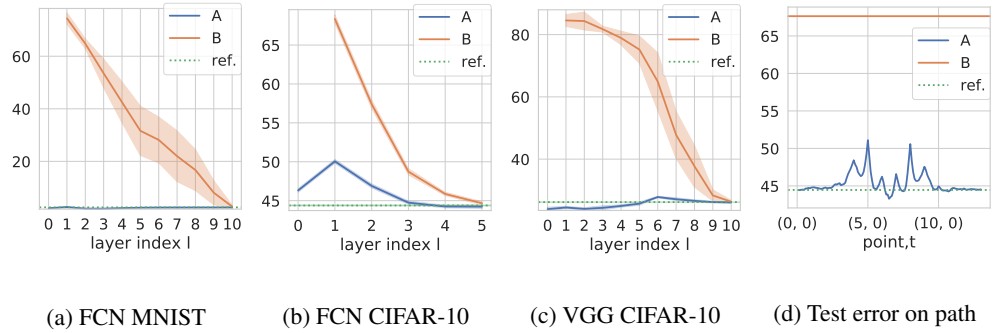

(a) FCN MNIST      (b) FCN CIFAR-10      (c) VGG CIFAR-10      (d) Test error on path

Figure 6: Test error (in %) for the experiment in Figure 3.

## C   Training Details

All models were trained from scratch on MNIST and CIFAR-10. The hyperparameters for training the original networks $\theta'$ (O) and for Experiment (A) can be found in Table 1. They were manually chosen in such a way that SGD achieves a reasonable convergence speed. All experiments were run on Nvidia GPUs Tesla V100. The implementation can be found on Github[3]. The original models are trained until 0 training error is reached. For Experiment (A), training is done for at least 100 epochs, and then it is stopped when 0 error or a maximum of 400 epochs is reached. Moreover, subnetworks $\xi_l$ with larger depth (i.e. starting at a lower $l$) required more care and a lower learning rate than shallow ones. For VGG, we employ a learning rate scheduler. As reported in the table, we tweak the scheduler for certain values of $l$ (giving the deepest models) in the case $p = \frac{1}{4}$. For the varying-width experiment (Figure 4), the same learning rate is used for different widths to allow a comparison of results. The maximal learning rate that makes every model converge is kept.

**Pruning for VGG-11.** As in previous work [24], for CNN architectures like VGG-11, each convolutional feature map associated with a filter/kernel can be treated as a single neuron of a fully connected architecture. In this way, Experiment (B) can be done similarly to the case of fully connected networks. For Experiment (A), this means that the subnetworks $\xi_l$'s consist of a small fraction of the convolutional filters from each layer of the original network. All the max-pooling layers, as part of the original VGG-11 architecture, are kept unchanged.

---

[3]https://github.com/modeconnectivity/modeconnectivity

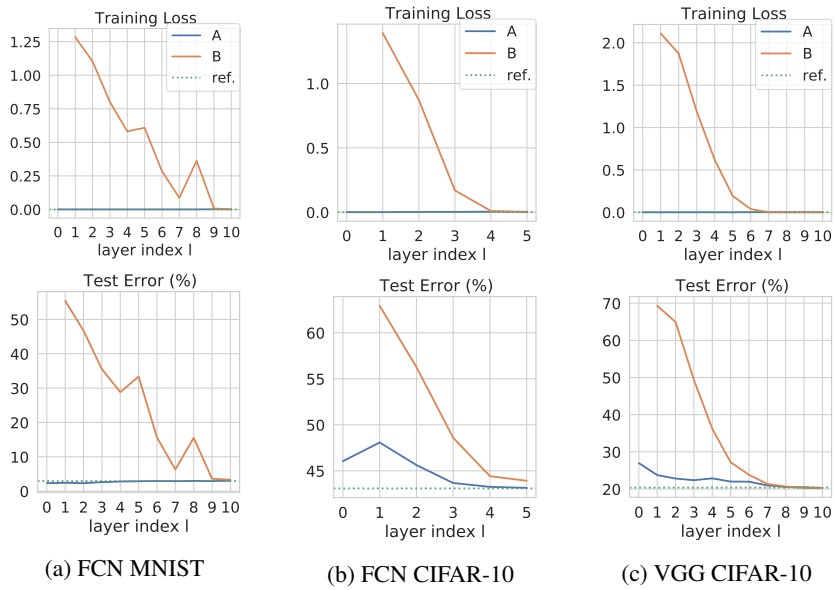

(a) FCN MNIST       (b) FCN CIFAR-10       (c) VGG CIFAR-10

Figure 7: Dropout ratio $p = \frac{1}{3}$. Training loss (top) and test error (bottom) for Experiments (A) and (B). The dotted green line is the originally trained model, used as a reference. We perform the experiments for only one trained model $\theta'$, and thus there is no confidence interval.

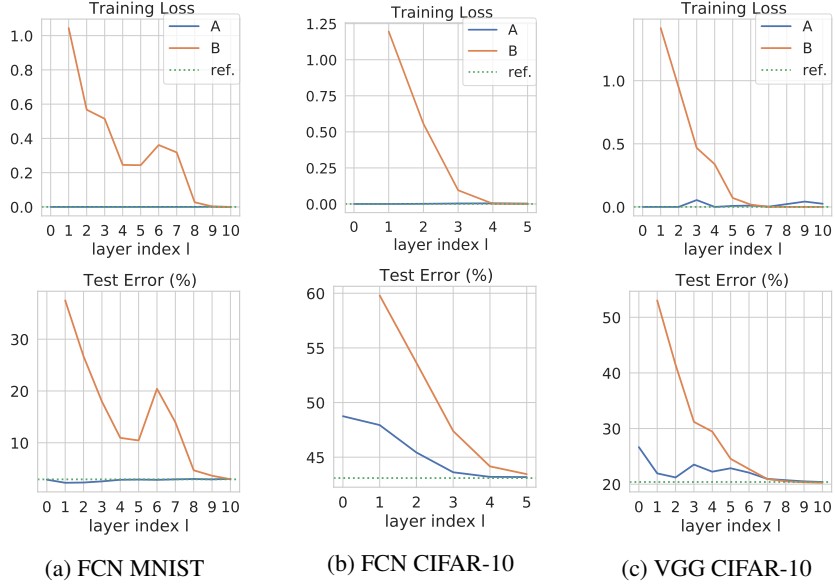

(a) FCN MNIST       (b) FCN CIFAR-10       (c) VGG CIFAR-10

Figure 8: Dropout ratio $p = \frac{1}{4}$. Training loss (top) and test error (bottom) for Experiments (A) and (B). The dotted green line is the originally trained model, used as a reference. We perform the experiments for only one trained model $\theta'$, and thus there is no confidence interval.

| Model | (O11) | (O3) | (A11) | (A3) |
|---|---|---|---|---|
| learning rate | $10^{-3}$ | $5 \cdot 10^{-3}$ | $2 \cdot 10^{-3} - 5 \cdot 10^{-3}$ | $5 \cdot 10^{-3}$ |
| momentum | 0.95 | — | — | — |

(a) FCN MNIST

| Model | (O6) | (O3) | (A6) | (A3), $l = 0$ | (A3), $l = 1, 2$ |
|---|---|---|---|---|---|
| learning rate | $10^{-3}$ | $5 \cdot 10^{-4}$ | $10^{-3} - 5 \cdot 10^{-3}$ | $10^{-3}$ | $5 \cdot 10^{-3}$ |
| momentum | 0.95 | — | — | — | — |

(b) FCN CIFAR-10

| Model | (O) | (A)* | (A), $p = \frac{1}{4}, l = 0$ | (A), $p = \frac{1}{4}, l = 1$ |
|---|---|---|---|---|
| learning rate | $5 \cdot 10^{-3}$ | — | — | — |
| momentum | 0.9 | — | — | — |
| weight decay | $5 \cdot 10^{-4}$ | — | — | — |
| scheduler | $\gamma = 0.5$ steps = 10 | $\gamma = 0.5$ steps = 20 | $\gamma = 0.1$ steps = 250 | $\gamma = 0.9$ mode = "plateau" |

* All of the cases except $p = \frac{1}{4}, l \in \{0, 1\}$

(c) VGG CIFAR-10

Table 1: Hyperparameters of the SGD optimizer for training the original model (O) and the subnetworks of Experiment (A). For each $K$, the models (O$K$) and (A$K$) refer to those subnetworks that have total number of layers $K$ (i.e. $K \in \{6, 11\}$ in Figure 3, and $K = 3$ in Figure 4). The index $l$ refers to the subnetwork $\xi_l$. Dashes stand for duplicate values from the left-most column.

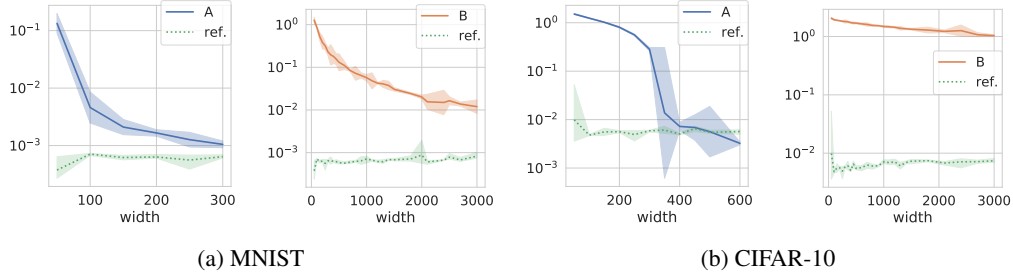

(a) MNIST      (b) CIFAR-10

Figure 9: Results of Figure 4 plotted with a logarithmic $y$-axis. Mean and 95% confidence intervals are depicted for 3 different trained networks $\theta'$.