# OpenReview forum: "When Are Solutions Connected in Deep Networks?"
_NeurIPS.cc/2021/Conference — NeurIPS 2021 Poster_

### Official Review · Reviewer_5bCi · 2021-07-02

**Rating:** 6
**Confidence:** 2

**Summary:**

This paper studies the connectivity of two parameters such as two SGD solutions of the deep neural networks. The general result subsumes several conditions including dropout stability, memorization capacity, and linear separable features. In particular, the theory shows that the connectivity occurs when the last two layers have $\Omega(\sqrt{N})$ neuron. The theory is validated by numerical experiments.

**Limitations And Societal Impact:**

The limitations are discussed in the paper. The potential negative societal impacts are not available, as this paper focuses on the mathematical theory for the solutions of deep networks.

**Main Review:**

Overall, I think the main message is interesting, and the presentation is clear. Since the main contribution claimed in this paper is on the milder overparametrization, the assumption (A1-b) on the feature quality deems further justifications. This assumption possibly implicitly put stringent requirements on the level of overparametrization of previous layers. The discussion might also help understand the difference between deep nets and two-layer nets where $\Omega(N)$ neurons are necessary.

**Time Spent Reviewing:**

5

---

> ### Author Response · Authors · 2021-08-09
> **Author Response**
>
> Thank you for your comments. Below, we elaborate on Assumption (A1-b) and we will include this discussion in the revision.
>
> Assumption (A1-b) appears to be rather mild. In fact, by taking $|\mathcal I_{L-1}^{(i)}|=(1-\delta)n_{L-1}$, the assumption holds as long as the loss does not change much after removing a $\delta$-fraction of the neurons in the last hidden layer. This is a much weaker condition than e.g. dropout stability, which requires that the loss does not change much after removing *half* the neurons at *every layer*. At the same time, we agree with the reviewer that it is an important next step to understand this condition (A1-b) in a more quantitative fashion, although we conjecture that having sub-linear widths at those layers would be sufficient.
> Let us conclude by mentioning that the numerical results of Section 6 show that our requirements (including Assumption (A1-b)) are satisfied by the solutions found by SGD in a variety of experimental settings.

---

> > ### Comment · Reviewer_5bCi · 2021-08-17
> > **Feedback**
> >
> > Thanks for the explanations. I see this is weaker than dropout stability, but the implication on the level of overparametrization is unknown. I think the message is more clear now.

---

> > > ### Author Response · Authors · 2021-08-17
> > > **Author Response**
> > >
> > > Let us add some further remarks about A1b that the reviewer asked.
> > >
> > > First, we think that the problem of removing condition A1b is more related to the worst-case analysis of the global connectivity of the landscape (sublevel sets), whereas the main motivation of our study here comes from the mode connectivity phenomenon (concerning the areas of the loss surface where SGD often land). In this last setting, we showed that, having more than 2 layers with sqrt(N) neurons or certain structures in the data, could already make a difference comapared to the previous setting (where the worst-case bound requires N+1 neurons).
> > >
> > > At the same time, we also want to remark that, the problem of analyzing the power/impact of depth on the optimization landscape of neural networks in general is a very challenging task, and we are not aware of any theoretical result so far in the literature on this front. In this regard, our paper made a solid step by drawing a formal connection between two properties of neural networks -- the memorization capacity (or expressivity) and the solution connectivity, which are currently both active research areas in deep learning. We hope that such a connection would lead to more insights and fruiful results in the future.
> > >
> > > Thanks again for your feedback. We are happy to discuss further any points that you might have.

---

### Official Review · Reviewer_S8SJ · 2021-07-13

**Rating:** 6
**Confidence:** 3

**Summary:**

The authors propose a theoretical explanation to the mode connectivity phenomena observed in neural networks. Specifically, it is shown that any two points in parameter space may be connected by a piece-wise linear path of low loss.  Technically this is done by assuming that there exists a tradeoff between feature quality and over parameterization at every layer. The authors note that this tradeoff assumption is milder than the dropout stability assumption made in previous work. Given this assumption, the authors improve upon the conditions for the existence of level set connectivity by requiring that the last two hidden layers only contain O(\sqrt{N}) neurons, instead of O(N).

**Limitations And Societal Impact:**

They are addressed

**Main Review:**

I am mainly concerned with the actual significance of the results, and with the writing/presentation in general.
As with the results themselves, i am a bit unsure as to how significant the results really are, given the strong assumption on the quality/over parameterization tradeoff. Concretely, Theorem 4.1 is proven by constructing sub networks connecting a set of neurons from each layer, to the output. The bound in Theorem 4.1 therefore depends on the error that these subnetworks can achieve (Eq 8), which crucially depend on the quality of the set of features chosen. It is not clear to me how significant this bound is given this assumption. (this is also true for the dropout stability assumption in prior work).   It appears to me that the main novelty of the paper is by formalizing a stronger assumption (which is eq 8 needs to be small) than dropout stability (the validity of both is not clear), and using it to get stronger results.

In addition, i find it hard to judge the impact of this paper due to the lack of discussion on why the subject matter is of any relevance. Besides it being intellectually intriguing to some, what is the significance of the findings in this paper, practically or otherwise?  To be clear, i feel this is more of a presentation issue than anything, and could be cleared out by the authors with some additional discussion.

Some additional comments:

- The authors mention that "This improvement comes at the cost of ensuring a weaker guarantee (connectivity of SGD solutions, as opposed to connectivity of all sublevel sets)". Can the authors expand on this further, how are the results in the paper specifically relevant for solutions found by SGD? Is the intention that the connecting path is found by SGD? This is not clear from the draft.

- While i appreciate figure 1 and the Proof techniques section, i feel that the readability of the proof of Theorem 4.1 could and should be improved. This is especially true since it is part of the main draft. For instance, i find it hard to parse the sentence ". Now, set the
incoming weights of the neurons ....  to the corresponding values given by...." right before Eq 13. This could be made much clearer by some additional figures similar to fig 1.

----------Post Rebuttal

The authors have addressed my concerns to an adequate degree, hence i have updated my score accordingly and recommend acceptance.


**Time Spent Reviewing:**

6

---

> ### Author Response · Authors · 2021-08-09
> **Author Response**
>
> Thank you for your comments: by incorporating the discussion below in the revision, we will improve the quality of our contribution.
>
> 1. *Comment: “I am mainly concerned with the actual significance of the results...”*
>
> **Response.** Let us clarify below the significance of our results.
>
> **(i) Improvement over prior work.** The surprising phenomenon of mode connectivity has been observed experimentally in several recent works [2, 12, 14, 18]: it is possible to connect solutions found by SGD via paths of low loss. This empirical observation has attracted the interest of the community looking for a theoretical explanation, namely, a characterization of the properties of the solutions which ensure the existence of a low-loss path connecting them. To this end, [27] shows that a dropout stability assumption suffices to ensure the connectivity. Here, we improve over prior work by providing a novel condition ensuring connectivity. Our condition is **provably weaker** than dropout stability, see Corollary 4.2, and this improvement is validated by the numerical experiments in Section 6: we are consistently able to find low-loss paths even for networks where the path constructed in earlier work (which relies on a dropout stability assumption) leads to a large loss. In this regard, our work has significantly improved the theoretical understanding of the mode connectivity phenomenon.
>
> **(ii) Impact on future research.** Our new condition trades off the quality of features with the overparameterization and, by doing so, it is able to capture the impact of depth on the optimization landscape. More specifically, it is known that, for a shallow network, having $N+1$ neurons ($N$ being the number of training samples) is necessary and sufficient for the sublevel sets of the loss to be connected [37]. Here, we show that, in order to find a low-loss path, it suffices to have two layers with $\Omega(\sqrt{N})$ neurons, provided that a mild condition on the features of the last layer is satisfied, see Corollary 4.3. Generally speaking, understanding the effect of depth on the optimization landscape is a fundamental open problem in the theory of deep learning, and our contribution makes a solid step towards this goal by drawing one of the first (formal) connections, to our knowledge, between the connectivity research and the memorization capacity research in deep learning.
>
> 2. *Comment: “Concretely, Theorem 4.1 is proven by constructing sub networks connecting a set of neurons from each layer, to the output. The bound in Theorem 4.1 therefore depends on the error that these subnetworks can achieve (Eq 8), which crucially depend on the quality of the set of features chosen. It is not clear to me how significant this bound is given this assumption. (this is also true for the dropout stability assumption in prior work)”*
>
> **Response.** Let us emphasize that our bounds are significant in several ways.
>
> First, it allows us to ignore the quality of features chosen at all the layers (except the last one) provided that the network is mildly over-parameterized (e.g. by having $\sqrt(N)$ neurons at the last two hidden layers). This is shown in our Corollary 4.3, where the quantity in (Eq 8) is only required to be small at the last hidden layer. This is in contrast with dropout stability, where (Eq 8) is required to be small for all the layers $l$. In practice, after training the network with SGD, the features at the last hidden layer are often good enough (e.g. they become linearly separable), and hence the relevance of our condition.
>
> Second, our bound allows us to show another *extreme* result, namely, if the trained network has already good enough (e.g. linearly separable) features at all the layers, then no over-parameterization is needed to show the connectivity.
>
> Taken together, by specializing our bounds to a variety of settings, the message of the paper is that there is a trade-off between the quality of features and over-parameterization of the network, which governs the mode connectivity phenomenon; and understanding this problem could be key to understanding the impact of depth on the optimization landscape of deep neural networks. Our formal discussion of the bounds, the potential impact of the data structure and the benefit of having more than 2 layers on the geometry of the loss landscape have provided a first step towards achieving this ambitious goal.
>
> 3. *Comment: “It appears to me that the main novelty of the paper is by formalizing a stronger assumption (which is eq 8 needs to be small) than dropout stability (the validity of both is not clear), and using it to get stronger results.”*
>
> **Response.** We believe there is a misunderstanding here: we do **not** make a stronger assumption than dropout stability to prove the result. In contrast, we prove in Corollary 4.2 that the dropout stability condition (as used in the prior works) *implies* our sufficient conditions of Theorem 4.1, and hence the connectivity result follows. In other words, our assumptions are theoretically **weaker** than dropout stability, and this is also numerically verified in our experiments.
>
> 4. *Comment: “Can the authors expand on this further, how are the results in the paper specifically relevant for solutions found by SGD?”*
>
> **Response.** Let us first recall that the connectivity of sublevel sets is a global property of the landscape which concerns *all* the points. In particular, it implies two properties: (i) from any point in parameter space, there is a continuous descent path towards a global minimum; and (ii) all the global minima are connected. In this paper, we are mainly interested in the connectivity of a *subset* of solutions in parameter space. Although our bound in Theorem 4.1 applies to any pair of points in the parameter space, the main interest and motivation for our work stems from the fact that in practice the solutions found by SGD can be often connected by a path of low cost. Thus, we use this theorem as a tool to (i) characterize the properties of a subset of points in parameter space which make them connected, and most importantly (ii) show that this characterization captures the set of solutions found by SGD. Our experiments in Section 6 confirm the relevance of our theory to SGD.
>
> 5. *Comment: “Is the intention that the connecting path is found by SGD?”*
>
> **Response.** In our proof, we provide an explicit construction for the low-loss path, and this requires the $\min\max\inf$ term in (Eq 4) in our Theorem 4.1 to be small. To solve this, we use the characterization in (Eq 6). Concretely, as discussed in our experiments in lines 314-317, we showed that it suffices to bound the quantities in (Eq 14), and for that we used an independent SGD routine to optimize the parameters of these subnetworks. In general, the existence of low-cost solutions in the training of these subnets can be theoretically guaranteed by exploiting results from memorization capacity (expressivity) of deep neural nets, and we showed one of these examples in Corollary 4.3.
>
> Finally, we remark that there could exist possibly many paths which connect two distinct solutions in parameter space along which the loss does not increase much, and our proof provides a way to construct one of them.
>
> 6. *Comment: “For instance, i find it hard to parse the sentence ". Now, set the incoming weights of the neurons .... to the corresponding values given by...”*
>
> **Response.** As mentioned in lines 232-234, whenever we say “set some parameter $w$ from $a$ to $b$”, we simply mean to change the current value of $w$ from $a$ to $b$ by following a direct path: $w(\lambda) = (1−\lambda)\cdot a + \lambda\cdot b$, where $\lambda\in[0,1]$. All other parameters, if not mentioned otherwise, should stay fixed. We hope this is clear now. Otherwise, please let us know.
>
> 7. *Comment: “This could be made much clearer by some additional figures similar to fig 1.”*
>
> **Response.** Thanks for this suggestion! As also mentioned in the response to reviewer 4Z11, we will provide supporting figures in the revision.

---

### Official Review · Reviewer_Yt3T · 2021-07-15

**Rating:** 7
**Confidence:** 4

**Summary:**

The paper introduces an upgrade of the existing research on the existence of low-loss paths connecting minima (or random points) on the loss surface of a neural network. The main idea is to bound the loss of the points along the path by the loss of the sparse subnetworks of the original network. The technique is demonstrated to be more widely applicable as compared to the existing dropout-stability technique. Moreover, the connection to memorization ability is shown with smaller requirements to the width of the layers, which is also demonstrated in the experiments (by the price of considering only SGD solutions and not the full parameter space). The theory gives an insight on the trade-off between quality of features and overparametrization of the layers.
The empirical evaluation demonstrates that the proposed approach can find connection path on each level and with significantly better approximation than the dropout stability technique. Also it requires less overparametrization for precise estimations.

**Limitations And Societal Impact:**

The authors included a detailed discussion of limitations, all of which seem to be viable.

As a theoretical paper it does not have any direct societal impact.

**Main Review:**

The paper is clearly written and provides the proofs for all the presented results. Empirical evaluation demonstrates the validity of the proposed technique also showing the advantages compared to the current state-of-the-art.

Some minor points can improve the understanding of the interesting details:
- Clearer discussion on memorization, including the notion of memorization used and how it is connecting to the low-loss-paths
- Clearer discussion on why only SGD solutions should be considered in the theory and what are the possible problems with it (will the parameter configurations built for the low-loss-path always an SGD solution for example?)

Possibly there is a typo in eq.4 (second max in RHS)

Overall, the theory is clear and well explained, the empirical results are good, so my suggestion would be to accept the paper for the conference.

---
I am satisfied with the rebuttal discussion and not changing my score.

**Time Spent Reviewing:**

4

---

> ### Author Response · Authors · 2021-08-09
> **Author Response**
>
> Thanks for your thoughtful comments. We address below your minor points and we plan to incorporate these discussions in the revision.
>
> 1. *Comment: “Clearer discussion on memorization, including the notion of memorization used and how it is connecting to the low-loss-paths”*
>
> **Response.** In our proof, the notion of memorization capacity (also known as finite sample expressivity) refers to the ability of the network in fitting a given finite dataset, or more specifically, the smallest number of neurons (or parameters) that the network needs in order to fit the dataset. Let us highlight that this has nothing to do with gradient descent algorithms or training, and it is a pure property of the architecture itself. The study of the memorization capacity of neural networks is a classical topic, dating back to the work of Cover in the 1960’s [11] and Baum in the 1980’s [5], and it has seen a recent revival, see [4, 8, 19, 35, 48, 53]. One of the contributions of this paper is to connect this notion of memorization capacity to the connectivity of the landscape: Corollary 4.3 shows that, in order to find a path with low loss between two distinct solutions, we can assume little from the features as long as the last two layers are sufficiently overparameterized. The basic idea is the following. First, if we remove $\Omega(\sqrt{N})$ neurons ($N$ being the number of training samples) from the last two hidden layers, the loss does not change significantly. Second, the subnetworks containing $\Omega(\sqrt{N})$ neurons in the last two layers are sufficiently overparameterized to memorize the whole training data set. Hence, we can use those subnetworks with zero training loss to construct the low loss path connecting the two solutions, with the recipe provided by Theorem 4.1.
>
> 2. *Comment: “Clearer discussion on why only SGD solutions should be considered in the theory and what are the possible problems with it (will the parameter configurations built for the low-loss-path always an SGD solution for example?)”*
>
> **Response.** SGD solutions provide the main motivation for this paper, since the phenomenon of mode connectivity was first discovered for neural networks trained via SGD (or its variants). However, let us highlight that our bound (4) on the loss provided in Theorem 4.1 holds for *any* pair of solutions, not necessarily those obtained from SGD algorithms. Given the aforementioned motivation, in our numerical experiments, we show that our upper bound on the loss of the connecting path leads to excellent results, when evaluated on solutions coming from SGD.
>
> 3. *Comment: “Possibly there is a typo in eq.4 (second max in RHS)”*
>
> **Response.** The second max on the RHS of Eq. 4 is intended to be taken between the two components in the open bracket (i.e., $\Phi(\theta’)$ and the $\min\max\inf$), thus it has no argument written below. We hope to have clarified this point now. In case Eq. 4 is still not clear, we are happy to elaborate more.

---

> > ### Comment · Reviewer_Yt3T · 2021-08-13
> > **Feedback**
> >
> > I thank the authors for their explanations and clarifications.
> >
> > 1 - So basically the main idea is that larger amount of neurons means larger memorization and the ability to apply the proposed technique depends on the amount of the neurons - do I understand it correctly?
> >
> > 2 - I guess this is written slightly confusing in the contributions part of the introduction. If the main theorem 4.1 already proves the existence of the low loss path for any two points, why the corollary for the technique introduces some restrictions?
> >
> > 3 - I understand now. I would suggest to think on the improvement of the readability of this formula (maybe breaking it in parts and giving separate notations can help).

---

> > > ### Author Response · Authors · 2021-08-17
> > > **Author Response**
> > >
> > > 1. Yes, the reviewer is correct. A larger amount of neurons means larger memorization capacity, which in turn means that it is easier to obtain subnetworks with low loss. More specifically, Corollary 4.3 makes this statement quantitative: under mild restrictions on the features, having the last two layers with $\Omega(\sqrt{N})$ neurons suffices to ensure the connectivity.
> > >
> > > 2. Thank you for this comment, we will incorporate the discussion above into the contributions part, in order to make our message more clear. Let us additionally comment on the role of the various corollaries. Theorem 4.1 provides an upper bound on the loss along the path connecting two given points $\theta_0$ and $\theta_1$.
> > > We note that the value of this upper bound depends on the choice of these two points in parameter space. In this regard, our subsequent corollaries show that, the upper bounds (as provided by Theorem 4.1) are guaranteed to be small in several interesting cases: (1) dropout stable solutions in Corollary 4.2, (2) networks with sufficient (sublinear) over-parameterization in Corollary 4.3, and (3) linearly separable features in Corollary 4.4. In other words, we regard the various corollaries as specific instantiations of the general framework provided by Theorem 4.1.
> > >
> > > 3. Thank you for this suggestion: in the revision, we will add a comment right below Eq. 4 in order to clarify how the second maximum is computed.

---

### Official Review · Reviewer_4Z11 · 2021-07-18

**Rating:** 7
**Confidence:** 3

**Summary:**

1. The authors prove a mild condition for mode connectivity for deep nets. The proof is a natural generalization of (Kuditipudi et al., 2019) and the condition required in this paper is provably weaker than the dropout stability. The authors also demonstrate in experiments that their condition can still hold when dropout stability fails.

2. The author provides a nice conceptual connection between their results and the memorization capacity of neural nets. In detail, they show if the product of widths of the last two layers is $\Omega(N)$, then solutions are connected via a low-loss path with minimal assumptions.

**Limitations And Societal Impact:**

Yes, in Sec 7.

**Main Review:**

This paper made a nice observation that the proof technique in (Kuditipudi et al., 2019) could be generalized in the following sense: for a fixed subset of the neurons per layer, to find a low-loss path, one doesn't have to rescale the original weights. Instead, it's possible to optimize weights in all further layers, thus further reducing the loss. Despite the simplicity of the idea, this new construction actually gives paths of much lower cost, as demonstrated in Figure 2 and 3.

This paper is clearly written and I enjoyed reading this paper. The setting of the experiments is clearly stated and I checked the proof of the main theorem and it sounds correct to me. The topic of mode-connectivity in deep learning is closely related to the interest of Neurips community. As the authors remarked, the mystery of mode-connectivity is still unsolved because we don't know why SGD find networks satisfying the assumption or if this is in general true for deep nets. However, I think this paper makes a solid step towards understanding mode-connectivity. Thus I suggest accepting this paper.

Other comments:
I really appreciate the authors' effort in making Figure 1, which helps readers to understand the notation of subnetwork used in this paper. Similarly, I think the paper will benefit from making several graphical demonstrations for the proof sketch, e.g., the operations you take to move from the right to the right subfigure in Figure 1.

**Time Spent Reviewing:**

4

---

> ### Author Response · Authors · 2021-08-09
> **Author Response**
>
> We thank the reviewer for the positive feedback. Thanks for the suggestion concerning the graphical demonstration of the proof sketch. We will include it in the final version.

---

### Decision · Program_Chairs · 2021-09-27

**Decision:**

Accept (Poster)

**Comment:**

The primary contribution of this work is in proving results similar to existing ones (connectedness of solutions in neural network training landscapes), but under milder assumptions.  Reviewers generally agreed that while the work is incremental by nature, it is technically sound and of interest to the community.  The AC sides with this view and recommends acceptance.